# Quality of antenatal care and its potential impacts on delivery services and postnatal care compliance among reproductive women in Bangladesh: A situation analysis from the Bangladesh Demographic and Health Survey 2017

Mehejabin Nurunnahar[1]*, M. Pear Hossain[2,3], Tahmidul Haque[4], S. M. Rokonuzzaman[4,5], Susmita Dey Pinky[6], Rumpa Kairy[4], Tahrima Mohsin Mohona[4], Abdus Sobhan[7], Most. Hafeza Khatun[8], Abu Yousuf Md Abdullah[9], Md Shahjahan Siraj[4]

**1** Institute of Epidemiology, Disease Control and Research (IEDCR), Dhaka, Bangladesh, **2** WHO Collaborating Centre for Infectious Disease Epidemiology and Control, School of Public Health, Li Ka Shing Faculty of Medicine, The University of Hong Kong, Pok Fu Lam, Hong Kong SAR, **3** Laboratory of Data Discovery for Health Limited, Hong Kong Science Park, Pak Shek Kok, Hong Kong SAR, **4** International Centre for Diarrhoeal Disease Research, Bangladesh (icddr,b), Mohakhali, Dhaka Bangladesh, **5** Arnold School of Public Health, University of South Carolina, Columbia, South Carolina, United States of America, **6** Imperial College Healthcare NHS Trust, London, United Kingdom, **7** Chief Economist's Unit, Bangladesh Bank, Dhaka, Bangladesh, **8** Binary data lab, Dhaka, Bangladesh, **9** School of Planning, University of Waterloo, Ontario, Canada

\* nishaju99@gmail.com

## Abstract

### Background

Ensuring quality antenatal care (ANC) and postnatal care (PNC) is crucial for reducing maternal and neonatal mortality rates. However, there are gaps in assessing the quality of ANC, leading to the proposal of standards by the World Health Organization. The study aims to examine the impact of quality ANC on delivery services and PNC compliance in Bangladesh using data from the Bangladesh Demographic and Health Survey (BDHS), providing insights for policymakers to improve maternal and neonatal health outcomes.

### Methods

This study used data from 2017 Bangladesh Demographic and Health Survey (BDHS) to investigate the impact of quality antenatal care (qANC) on delivery services and PNC in Bangladesh. The study population included ever-married women aged 15–49 years who had experienced a recent pregnancy. The analysis assessed the relationship between qANC and facility delivery, skilled birth attendant (SBA)-assisted delivery, and PNC services within 48 hours of delivery. The study employed

**Data availability statement:** All relevant data are within the paper and its Supporting Information files.

**Funding:** The author(s) received no specific funding for this work.

**Competing interests:** The authors have declared that no competing interests exist.

**Abbreviations:** ANC, Antenatal Care; ARR, Adjusted Risk Ratio; BBS, Bangladesh Bureau of Statistics; BDHS, Bangladesh Demographic and Health Survey; EA, Enumeration Area; MMR, Maternal Mortality Ratio; MTP, Medically Trained Provider; NMR, Neonatal Mortality Rate; PNC, Postnatal Care; PSU, Primary Sampling Unit; qANC, Quality Antenatal Care; SBA, Skilled Birth Attendant; UN, United Nations; WHO, World Health Organization

a two-stage stratified cluster sampling design, and data analysis was conducted using generalized linear models and considered various demographic and socioeconomic factors.

## Results

Key findings include a low rate of qANC services (18%), with pregnancy-related counseling being the lowest component. About 82% received at least one ANC visit, but only 18.3% received a quality visit. Higher compliance with facility delivery (ARR: 1.3; 95% CI: 1.27–1.41), SBA-conducted delivery (ARR: 1.3; 95% CI: 1.24–1.35), and PNC services for both mother (ARR: 1.3; 95% CI: 1.24–1.35) and child (ARR: 1.3; 95% CI: 1.23–1.35) within 48 hours were observed when quality ANC was received. Factors such as completing secondary education, engaging in skilled/unskilled manual labor and higher wealth quintile were associated with better delivery and post-delivery outcomes.

## Conclusion

Ensuring qANC and expanding PNC service use remain challenging in Bangladesh. Increasing the provision of qANC is crucial, as it is associated with higher adherence to PNC.

## Introduction

A crucial component of ensuring maternal and newborn health and wellbeing is providing proper care before, during, and after pregnancy. According to a systematic analysis conducted by United Nations (UN) Maternal Mortality Estimation Inter-Agency Group (MMEIG), during 2015 almost 830 women died per day worldwide due to difficulties in pregnancy or childbirth, and roughly 99% of these deaths occurred in developing countries [1,2]. World Health Organization (WHO) estimates reported a staggering 2,95,000 deaths during 2017 from pregnancy related complications, where Sub Saharan Africa and Southern Asia accounted for 86% of the deaths, with Southern Asia having a toll of 20% (58,000 deaths) [3]. India alone stood for around 12% of the global maternal deaths that year. Even by then and 2020, death tally per day was around 810 and 800 respectively, with the largest chunk of the burden falling upon again on lower middle income countries, meaning the situation has negligibly improved over the past few years [4]. However, ensuring quality antenatal care (ANC) and postnatal care (PNC) could have substantially reduced the risks during and after pregnancies and childbirth [1], saving hundreds of thousands of lives in these countries. According to some studies, quality ANC alone has the capacity to reduce maternal deaths by 20% on regular attendance [2,5].

Contrary to the past belief that the quality of ANC provided could be assessed by focusing on the number of contacts between women or newborns with the healthcare system, recent studies noted critical gaps in the measurement and reporting of

the quality of services through this approach [6–8]. Consequently, the WHO has proposed standards of care and measures for assessing the quality of maternal and newborn health care [6], highlighting the importance of both the quality and content of ANC services while assessing the quality of care [9]. With regard to quality of ANC, WHO has recommended a core set of essential contents including blood pressure and body weight measurement, blood hemoglobin and urine albumin tests, and counselling on dangers signs; which are integral to track maternal physical conditions during pregnancy [1].

In Bangladesh, the utilization of maternal health services such as facility delivery, delivery by a skilled birth attendant (SBA), and PNC for mothers and children have increased remarkably. Unfortunately, despite the strong efforts of the government, the maternal mortality ratio (MMR) decline has halted in recent years [10], and the Maternal Mortality Rate (MMR) is still very high (170 per 100,000 live births), as is the Neonatal Mortality Rate (NMR, 28 per 1000 live births). Evidences suggest that through increased coverage and quality of preconception, antenatal, intrapartum, and postnatal interventions, a considerable number of maternal and neonatal deaths, and also stillbirths can be forestalled [11,12]. There has been no recent attempt to examine the quality of current ANC practices in Bangladesh, nor has any study assessed the impacts of quality ANC on delivery services and PNC uptake among reproductive women, as past studies suggest that ANC experiences play a pivotal role in the uptake of PNC and thus, directly affect the maternal and neonatal health conditions [1,13–15]. Given this research gap, the present study attempts to identify the potential impact of quality ANC on delivery services and PNC compliance using data from the Bangladesh Demographic and Health Survey (BDHS). The findings would offer insights to policymakers about the different public health strategies to increase the quality of ANC, PNC and in greater aspect reduce neonatal and maternal mortality rate in Bangladesh.

## Methods

### Data source

This study analyzed the data from nationally represented dataset called BDHS of year 2017. It is a national level survey whose objective is to present the up-to-date estimates of demographic and health indicators specifically fertility, childhood mortality, maternal and child health, nutrition and newborn care. BDHS is a cross-sectional survey implemented by Mitra and Associates and conducted under the supervision of the National Institute of Population Research and Training (NIPROT) and the Ministry of Health and Family Welfare, Government of the People's Republic of Bangladesh. A detailed description of the survey design, methodologies, sample size and questionnaires have been provided in the final summary report of 2017–2018 BDHS [16].

### Study population and survey design

In BDHS, only ever-married women of aged 15–49 years were interviewed. We analyzed the data of those women who had a pregnancy in last 3–5 years, to assess the impact of qANC on delivery and PNC services for mother and child by medically trained healthcare providers within 48 hours of delivery. BDHS is based on a two-stage stratified sample of households where eight administrative divisions were treated as stratum. Here samples were collected covering urban (in two stages) and rural (in three stages) areas from all the administrative divisions in Bangladesh: Dhaka, Chottogram, Rajshahi, Sylhet, Barisal, Khulna, Rangpur and Mymensingh. A list of total 675 enumeration areas (EAs) (250 in urban areas and 425 in rural areas) was used as a sampling frame provided by the Bangladesh Bureau of Statistics (BBS) referring to the 2011 Population and Housing Census of the People's Republic of Bangladesh [17]. EAs' were also the primary sampling unit (PSU) covering an average of 120 households. During the first stage of sampling, rural wards were chosen, then PSUs, and finally families were chosen from PSUs. In urban areas, the PSUs method was used to choose wards, and one EA was chosen from each PSU. Then, in the second sampling stage, systematic sampling was implemented to select an average of 30 households per EA to provide statistically reliable estimates of the key demographic and health variables for the whole country, for urban and rural areas separately and for each administrative division.

 

## Dependent variables

The main dependent variables of our study were qANC, facility delivery, an SBA conducting the delivery and PNC services for both mother and child within 48 hours of delivery. According to the 2017–2018 BDHS definition, when a mother has received all the five core components of ANC such as measurement of blood pressure, measurement of weight, blood test for hemoglobin, urine test for albumin, and counselling on maternal danger signs during the antenatal visits and attending minimum of four ANC visits with at least one visit conducted by a medically trained provider irrespective of pregnancy trimester; she was considered to receive a qANC [16]. SBA delivery specifically refers to births attended by a skilled birth attendant, irrespective of the delivery location, whereas facility delivery denotes births that occur within a healthcare facility. Additionally, PNC refers to the care provided to both the mother and child following delivery.

## Explanatory variables

The explanatory variables included in our study were selected based on previous literatures reporting the quality of maternal and newborn care services and also the structure of BDHS reports. Participants (ever married women of 15–49 years age who experienced a pregnancy within last 3–5 years) reported their age in years (categorized as <19 years, 20–29 years, 30–39 years, 40–49 years), education level (categorized as no education, primary, secondary incomplete, secondary complete/higher), residence (urban, rural), division (Dhaka, Chottogram, Rajshahi, Sylhet, Barisal, Khulna, Rangpur, Mymensingh), wealth quintile, occupational status of the participants and their husbands (not working, agriculture, skilled/unskilled manual, services/sales) and birth order (1, 2–3, 4). Principal component analysis was conducted using data obtained on household construction materials and housing characteristics (i.e., source of water, sanitation facility, housing structure) along with ownership of durable assets to determine the wealth status of households for further determination of socioeconomic status; then the wealth status was stratified into quintiles (richest, richer, middle, poorer, poorest).

## Data analysis

Women aged 15–49 years and having at least one live birth in last 3–5 years were considered as the study population. The number of ever-married women in BDHS 2017 was 47,828. For the analysis, continuous variables such as mother's age, education, parity and birth order were converted into categorical variables. The main outcomes of the study were qANC, facility delivery, SBA conducting the delivery, and PNC services for both mother and child within 48 hours of delivery. The ANC components assessed in our study were based on the definition outlined in the 2017–2018 BDHS. In this study, we examined the effect of qANC on delivery care and receiving PNC from a medically trained provider (MTP) within 48 hours after the delivery.

Descriptive statistics were used to assess the sociodemographic characteristics of the targeted population. To evaluate the impact of qANC on PNC and delivery care, generalized linear model with binomial family and log link was fitted. Backward elimination was used to identify significant determinants and account for confounding factors, including variables with a p-value ≤ 0.20 in the adjusted model. Adjusted risk ratio (ARR) with 95% confidence intervals (CI) were calculated, and associations were considered as statistically significant at p < 0.05. Based on the previous studies, factors such as, mother's age, education, occupation, husband's occupation, birth order, and wealth quintile were considered as potential confounders. The data were analyzed using Stata version 14.0 (StataCorp, 2015; Stata Statistical Software: Release 14, College Station, TX: StataCorp LP).

## Ethical consideration

This study utilized data from the 2017 BDHS, which is publicly available and de-identified. Consequently, no additional ethical approval from any institution was required. The BDHS reports provide comprehensive details on the ethical considerations and procedures followed during the data collection process. As such, this study was exempt from ethical review approval and did not require participant consent, in line with guidelines for using publicly available de-identified data.

## Results

### Demographic characteristics

A general overview of the background characteristics of the study participants included in this study is presented in Table 1. A total of 47,828 ever-married women aged 15–49 years who had at least one live birth within the last 3–5 years were included in the analysis using data from the BDHS 2017 dataset. The mean age of the mothers in the analyzed sample was 24.9+5.6 years. A majority (61%) of the participants belonged to the 20–29 years' age group, while only 1% were aged 40 years or above. A significant number (40%) of mothers had started high school but could not complete their secondary education. In contrast, about 8% of the mothers had not received any formal education. A considerable number of mothers (61%) were unemployed, and another 27% were engaged in agricultural work. Only 3% of the participants were engaged in any service/sales-related work. On closer inspection of the occupations of the partners, a majority (58%) of the husbands were found to be engaged in skilled/unskilled manual labor, with only 8% involved in agriculture (Table 1).

About half of the mothers had 2–3 birth parity, and around 12% had a parity of more than 4. Over one-fourth (26%) of the mothers were urban residents, and the remaining (73%) lived in rural areas. The mothers did not have any distinct differences in terms of wealth quintiles (Table 1).

### Quality of ANC

The quality of ANC services that the mothers received, as analyzed based on the five core ANC components, is reported in Fig 1. The pregnancy-related counseling during ANCs was found to be lower than other components. Only two-fifth (40%) of the mothers had received counseling on pregnancy-related danger signs, whereas about two-thirds (66% and 73%, respectively) had blood and urine tests done. However, majority of the participants had their weight and blood pressure (89% and 94%, respectively) measured during ANCs. The overall qANC service provision rate was only 18% (Fig 1).

### Distribution of ANC exposure, delivery and post-delivery service outcomes

The results suggest that only 18% of the mothers did not receive an ANC during their pregnancies. Overall, a high proportion (82%) of the study participants were found to have received at least one ANC from an MTP. However, less than half (47%) of the mothers had 4+ ANC visits during their pregnancies.

Less than one-fifth (18.3%) of the mothers had obtained a quality visit (MTP performed all five core components). There was no significant differentiation among the mothers in terms of having a facility delivery, an SBA conducting the delivery, and receiving PNC services (mother and child) within the 48 hours after birth (Table 2).

### Distribution of delivery and PNC service compliances

Table 3 shows the distribution of delivery and PNC service compliances based on ANC visits during pregnancies and the background characteristics of the surveyed mothers. The results suggest that higher compliances to facility delivery (77.9%), delivery by an SBA (80.9%), and PNC for both mother and child within 48 hours after delivery (81%) could be achieved when qANC could be ensured to mothers during pregnancies. Additionally, a considerably higher percentage of facility delivery, delivery by an SBA, and PNC services were observed amongst mothers who had either completed their secondary education, were engaged in skilled/unskilled manual labors, and belonged to the richest wealth quintile, which likely enhances healthcare awareness, accessibility, and service uptake in these groups. These three outcomes were also highly prominent amongst mothers from Dhaka and Khulna divisions and less distinct in mothers from Barisal, Mymensingh, and Sylhet (Table 3).

### Association between demographic characteristics and outcome variables

Table 4 shows the adjusted multivariable logistic regression model for association between qANC and outcome variables (facility delivery, delivery by an SBA, and PNC services).

**Table 1. Background characteristics of women aged 15-49 who had a live birth in the BDHS 2017 survey.**

| Background characteristics | Number of women | % |
|---|---|---|
| **Mother's age** | | |
| <19y | 904 | 17.9 |
| 20-29y | 3086 | 61.1 |
| 30-39y | 1006 | 19.9 |
| 40-49y | 55 | 1.1 |
| **Mother's education** | | |
| No education | 390 | 7.7 |
| Primary | 1502 | 29.7 |
| Secondary education incomplete | 2031 | 40.2 |
| Secondary/higher education complete | 1129 | 22.3 |
| **Mother's occupation** | | |
| Unemployed | 3104 | 61.5 |
| Agricultural | 1365 | 27 |
| Manual labor (skilled/unskilled) | 411 | 8.1 |
| Service/sales | 171 | 3.4 |
| **Husband's occupation** | | |
| Not working | 666 | 13.2 |
| Agricultural | 428 | 8.5 |
| Manual labor (skilled/unskilled) | 2928 | 58 |
| Service/sales | 1030 | 20.4 |
| **Birth order** | | |
| 1 | 1931 | 38.2 |
| 2 - 3 | 2503 | 49.6 |
| 4+ | 617 | 12.2 |
| **Residence** | | |
| Urban | 1356 | 26.8 |
| Rural | 3695 | 73.2 |
| **Division** | | |
| Barisal | 288 | 5.7 |
| Chittagong | 1071 | 21.2 |
| Dhaka | 1293 | 25.6 |
| Khulna | 464 | 9.2 |
| Mymensingh | 431 | 8.5 |
| Rajshahi | 587 | 11.6 |
| Rangpur | 534 | 10.6 |
| Sylhet | 383 | 7.6 |
| **Wealth quintile** | | |
| Poorest | 1042 | 20.6 |
| Poor | 1036 | 20.5 |
| Middle-class | 969 | 19.2 |
| Rich | 1018 | 20.2 |
| Richest | 986 | 19.5 |

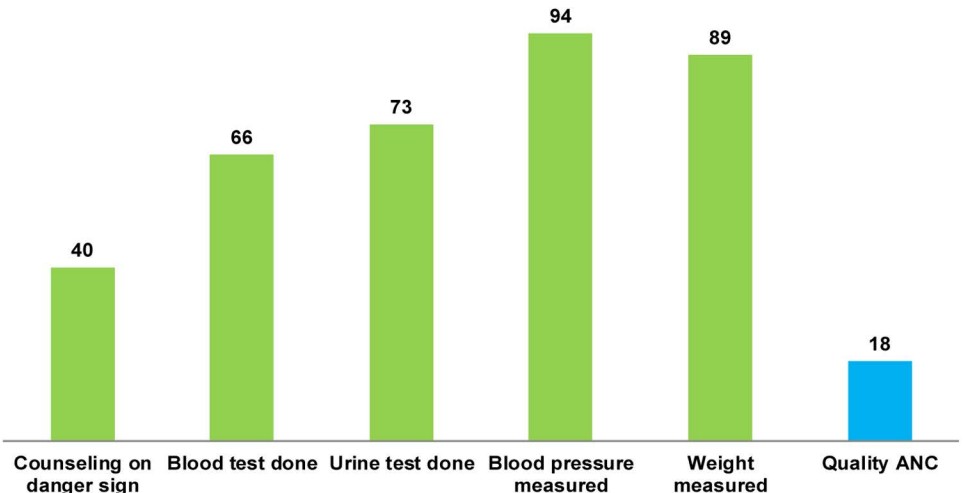

**Fig 1. The five components of quality ANC as received by women aged 15-49 years who had a live birth in the BDHS 2017 survey, (%).**

**Table 2. The background characteristics of ANC visits, nature of deliveries, and PNC services received by the study participants.**

| Background characteristics | Number of women | % |
|---|---|---|
| **Any ANC by MTP** | | |
| No | 914 | 18.1 |
| **Yes** | 4137 | 81.9 |
| **Number of ANC** | | |
| None | 405 | 8 |
| 1 | 661 | 13.1 |
| 2 | 830 | 16.4 |
| 3 | 781 | 15.5 |
| **4+** | 2374 | 47 |
| **Quality of ANC** | | |
| No | 4093 | 81.7 |
| Yes | 919 | 18.3 |
| **Facility Delivery** | | |
| No | 2533 | 50.1 |
| Yes | 2519 | 49.9 |
| **SBA** | | |
| No | 2379 | 47.1 |
| Yes | 2672 | 52.9 |
| **PNC for mother within 48 hours** | | |
| No | 2354 | 46.6 |
| Yes | 2697 | 53.4 |
| **PNC for child within 48 hours** | | |
| No | 2349 | 46.5 |
| Yes | 2702 | 53.5 |

**Table 3. Distribution of outcomes by ANC exposure and background characteristics, n (%).**

| Background characteristics | Facility Delivery | Delivery by SBA | PNC for mother within 48 hours | PNC for child within 48 hours |
|---|---|---|---|---|
| Quality ANC | | | | |
| No | 1824(43.9) | 1950(46.8) | 1975(47.5) | 1979(47.6) |
| Yes | 695(77.9) | 722(80.9) | 722(80.9) | 722(81) |
| **Mother's age** | | | | |
| <19y | 461(50.9) | 495(54.7) | 506(55.9) | 502(55.5) |
| 20-29y | 1559(50.5) | 1649(53.4) | 1653(53.6) | 1665(54) |
| 30-39y | 474(47.1) | 500(49.6) | 510(50.6) | 506(50.3) |
| 40-49y | 26(47.1) | 29(53.1) | 29(53.1) | 29(53.1) |
| **Mother's education** | | | | |
| No education (0 years) | 105(26.8) | 115(29.4) | 119(30.5) | 118(30.3) |
| Primary education (1–5 years) | 497(33.1) | 522(34.8) | 531(35.4) | 539(35.9) |
| Secondary incomplete (6–9 years) | 1048(51.6) | 1132(55.7) | 1149(56.6) | 1145(56.4) |
| Secondary complete or higher (10+ years) | 869(77) | 904(80) | 898(79.5) | 901(79.8) |
| **Mother's occupation** | | | | |
| Not working | 1713(55.2) | 1802(58.1) | 1814(58.4) | 1815(58.5) |
| Agricultural | 506(37.1) | 550(40.3) | 558(40.9) | 560(41) |
| Skilled/unskilled manual | 225(54.8) | 240(58.5) | 242(58.9) | 243(59.2) |
| Service/sales | 75(43.9) | 80(46.5) | 84(49) | 84(48.9) |
| **Husband's occupation** | | | | |
| Not working | 274(41.2) | 290(43.6) | 297(44.6) | 295(44.3) |
| Agricultural | 115(26.9) | 125(29.2) | 126(29.4) | 128(29.9) |
| Skilled/unskilled manual | 1532(52.3) | 1624(55.5) | 1638(56) | 1643(56.1) |
| Service/sales | 597(58) | 633(61.5) | 636(61.7) | 636(61.8) |
| **Birth order** | | | | |
| 1 | 1175(60.9) | 1247(64.6) | 1253(64.9) | 1251(64.8) |
| 2-3 | 1178(47) | 1240(49.5) | 1251(50) | 1261(50.4) |
| 4+ | 166(26.9) | 185(29.9) | 193(31.3) | 190(30.8) |
| **Residence** | | | | |
| Urban | 862(63.5) | 918(67.7) | 918(67.7) | 918(67.7) |
| Rural | 1657(44.8) | 1754(47.5) | 1779(48.1) | 1784(48.3) |
| **Division** | | | | |
| Barisal | 113(39.1) | 136(47.3) | 136(47.1) | 135(46.8) |
| Chittagong | 500(46.7) | 544(50.8) | 557(52) | 556(51.9) |
| Dhaka | 743(57.4) | 780(60.3) | 780(60.3) | 783(60.5) |
| Khulna | 283(61) | 296(63.7) | 301(64.9) | 300(64.6) |
| Mymensingh | 167(38.8) | 178(41.2) | 179(41.6) | 180(41.7) |
| Rajshahi | 309(52.7) | 321(54.7) | 325(55.3) | 326(55.6) |
| Rangpur | 254(47.5) | 263(49.1) | 263(49.2) | 263(49.2) |
| Sylhet | 150(39.2) | 154.5(40.3) | 157(40.9) | 160(41.6) |
| **Wealth quintile** | | | | |
| Poor | 274(26.3) | 289(27.8) | 297(28.5) | 295(28.3) |
| Poorest | 387(37.4) | 418(40.3) | 422(40.7) | 430(41.5) |
| Middle | 473(48.8) | 510(52.6) | 522(53.8) | 524(54) |
| Rich | 610(59.9) | 641(62.9) | 642(63.1) | 641(63) |
| Richest | 775(78.5) | 814(82.5) | 815(82.6) | 813(82.4) |

**Table 4. Association between outcomes and exposures, ARR (95% CI).**

| Background characteristics | Facility Delivery | Delivery by SBA | PNC for mother within 48 hours | PNC for child within 48 hours |
|---|---|---|---|---|
| **Quality ANC** | | | | |
| No | Ref | Ref | Ref | Ref |
| Yes | 1.3(1.27 - 1.41) | 1.3(1.24 - 1.36) | 1.3(1.24 - 1.35) | 1.3(1.23 - 1.35) |
| **Mother's age** | | | | |
| <19y | Ref | Ref | Ref | Ref |
| 20-29y | 1(0.92 - 1.02) | 1(0.92 - 0.99) | 1(0.92 - 1) | 1(0.92 - 1) |
| 30-39y | 0.9(0.9 - 1.03) | 0.9(0.9 - 1) | 1(0.91 - 1.01) | 1(0.9 - 1.02) |
| 40-49y | 0.8(0.72 - 1.11) | 0.9(0.75 - 1.07) | 0.9(0.76 - 1.08) | 0.9(0.76 - 1.09) |
| **Mother's education** | | | | |
| No education (0 years) | Ref | | Ref | Ref |
| Primary education (1–5 years) | 1.2(1.03 - 1.47) | | 1.2(0.98 - 1.36) | 1.2(1 - 1.39) |
| Secondary incomplete (6–9 years) | 1.8(1.52 - 2.13) | | 1.8(1.5 - 2.04) | 1.8(1.52 - 2.08) |
| Secondary complete or higher (10+ years) | 2.3(1.96 - 2.76) | | 2.2(1.88 - 2.57) | 2.2(1.92 - 2.62) |
| **Mother's occupation** | | | | |
| Not working | Ref | Ref | Ref | Ref |
| Agricultural | 0.9(0.84 - 0.98) | 0.9(0.87 - 1) | 0.8(0.77 - 0.88) | 0.8(0.77 - 0.88) |
| Skilled/unskilled manual | 1(0.93 - 1.06) | 1(0.94 - 1.05) | 1(0.89 - 1.02) | 1(0.9 - 1.02) |
| Service/sales | 0.8(0.72 - 0.99) | 0.8(0.71 - 0.96) | 1(0.86 - 1.14) | 1(0.85 - 1.13) |
| **Husband's occupation** | | | | |
| Not working | Ref | Ref | Ref | Ref |
| Agricultural | 0.8(0.7 - 0.99) | 0.9(0.72 - 1.01) | 0.8(0.71 - 1) | 0.9(0.73 - 1.02) |
| Skilled/unskilled manual | 1.1(0.99 - 1.2) | 1.1(1.01 - 1.21) | 1.1(1.01 - 1.2) | 1.1(1.02 - 1.21) |
| Service/sales | 1.1(1 - 1.21) | 1.1(1 - 1.21) | 1.1(1.01 - 1.21) | 1.1(1.03 - 1.23) |
| **Residence** | | | | |
| Urban | Ref | Ref | Ref | Ref |
| Rural | 0.9(0.97 - 1.07) | 0.9(0.96 - 1.04) | 1(0.96 - 1.05) | 1(0.96 - 1.04) |
| **Wealth quintile** | | | | |
| Poor | Ref | Ref | Ref | Ref |
| Poorest | 1.4(1.2 - 1.55) | 1.4(1.24 - 1.59) | 1.4(1.22 - 1.56) | 1.4(1.26 - 1.6) |
| Middle | 1.7(1.53 - 1.95) | 1.8(1.59 - 2) | 1.8(1.6 - 2.01) | 1.8(1.62 - 2.03) |
| Rich | 2(1.78 - 2.25) | 2(1.8 - 2.26) | 2(1.79 - 2.23) | 2(1.8 - 2.25) |
| Richest | 2.4(2.13 - 2.7) | 2.4(2.17 - 2.72) | 2.4(2.18 - 2.71) | 2.4(2.18 - 2.71) |

Receiving qANC was found significantly associated with facility delivery (ARR: 1.3; 95% CI: 1.27–1.41), delivery by an SBA (ARR: 1.3; 95% CI: 1.24–1.35) and PNC services for both mother (ARR: 1.3; 95% CI: 1.24–1.35) and child (ARR: 1.3; 95% CI: 1.23–1.35) within 48 hours. Mothers who completed their secondary education were more likely to receive facility delivery (ARR: 2.3; 95% CI: 1.96–2.76) and PNC services within 48 hours for them (ARR: 2.2; 95% CI: 1.88–2.57) and their child (ARR: 2.2; 95% CI: 1.92–2.62) compared to mothers with no formal education. Similarly, mothers belonging to the richest wealth quintile were more likely to receive facility delivery, delivery by an SBA and PNC services than those from the reference category (Table 4). These relationships highlight the reinforcing effect of qANC in improving maternal and child health outcomes among women from more advantaged socioeconomic backgrounds.

    

## Discussion

Despite the increases in ANC visits, only 18% of the mothers were receiving qANC during their pregnancies. Indications are clear that although the increasing trend in receiving ANC amongst pregnant women is encouraging, more focus is required to ensure the ANCs receipt conform to qANC standards. Previous studies found the positive outcomes among those who receive ANC services compared to those who don't. But qANC has more effect on facility delivery, SBA and PNC service uptake [13,16,18–20].

According to our findings, amongst the five qANC core components (blood pressure measurement, weight measurement, blood test for hemoglobin, urine test for albumin, and counseling on danger signs), the most commonly received components were blood pressure (94%) and body weight (89%) measurements. The least prominent qANC component was counseling on danger signs (40%), which is concerning as expectant mothers must be able to quickly recognize the danger symptoms to seek medical assistance on time. Hence, an increase in both the quantity and quality of counseling remains critical to ensuring qANC [18]. Different factors are responsible for poor counselling service during the ANC service. These include time constraints faced by healthcare providers, insufficient training of staff in effective counseling techniques, an overwhelming patient load, limited attention to the psychological aspects of pregnancy, and, sometimes a lack of awareness regarding the importance of comprehensive counseling [21,22].

Our analysis found a positive association between exposure to qANC and outcomes such as facility delivery, an SBA conducting the delivery, and compliance with PNC services for both mother and child within 48 hours. In the adjusted model, all outcomes were higher among the women who received qANC compared to those who did not. The ARR values of getting facility delivery, delivery by an SBA and PNC services within 48 hours were about 1.3 times among the women who got qANC. A longitudinal study in Bangladesh [19] also revealed that ANC had a positive impact on PNC utilization. Findings from systematic analysis and meta-analysis conducted in Ethiopia [15,20,23–25], as well as a meta-analysis conducted in low- and middle-income countries [25] concur with this finding. Our findings also align with a cross-sectional study derived from the Afghanistan Health Survey 2018, which underscores the potential of qANC to inform the design and implementation of health interventions aimed at optimizing service delivery during ANC visits [26]. Such insights are invaluable for tailoring maternal health programs to better meet the needs of pregnant women. Mothers' education level and belonging to the higher income quintiles were found to be significantly and positively associated with facility delivery, an SBA conducting the delivery, and compliance with PNC services. This conclusion of a strong influence of maternal education on optimal utilization of maternal care services is in agreement with previous studies in countries similar to Bangladesh [20,23]. A possible explanation for these observed results might be that educated mothers are more likely to understand the necessity of well-being during their maternity period and are aware of the resources that are best suited for ensuring a safe pregnancy. They are more aware of the available services. Additionally, income-generating activities can enhance a mother's decision-making capacity, position in families, and, thus, the ability to undertake health-beneficial decisions during pregnancies [14]. However, it is important to emphasize that these associations are also likely influenced by the qANC that these women receive. Women with higher education and income are more likely to access and benefit from qANC, which provides critical information and support for facility delivery, SBA-assisted delivery, and timely PNC services.

## Strengths and limitations

This is the first study evaluating the relationship between qANC and uptake of facility delivery, SBA and PNC service. As our study utilized data from a demographic survey, ensuring that the women included are nationally representative, which enhances the generalizability of the findings across diverse populations. Therefore, this comprehensive dataset serves as a reliable benchmark for future research. In addition, by addressing key maternal health indicators, our study contributes valuable insights to inform policy and interventions aimed at improving antenatal, delivery, and postnatal care services.

There are several constraints associated with this study. Recall bias may have occurred as women reported events and services from preceding three years, potentially resulting in an underreporting of the contents of ANC services. Furthermore, it is possible that certain women may have misidentified the health facilities or providers from which they received ANC services. Although we followed the BDHS definition and analyzed the five components based on the available BDHS data, interpreting quality ANC according to the WHO definition requires data on additional variables. This limitation could restrict the interpretation of our findings and may not capture the broader scope of maternal health care recommended by WHO.

## Conclusion

Although ANC services are being utilized more frequently, ensuring qANC and expanding PNC service utilization have remained a challenge for Bangladesh. Since qANC is associated with increased adherence to PNC, providing high-quality prenatal care to women may be the most effective method to promote PNC service utilization in the country.

## Supporting information

**S1 Table. Association between outcomes and exposures, unadjusted risk ratio (95% CI).**
(DOCX)

**S2 File. Raw data on quality ANC and its potential impacts.**
(XLSX)

## Acknowledgments

The authors want to sincerely thank all men and women participating in Bangladesh Demographic and Health Survey 2017.

## Author contributions

**Conceptualization:** Mehejabin Nurunnahar, Md Shahjahan Siraj.

**Data curation:** Mehejabin Nurunnahar, Md Shahjahan Siraj.

**Formal analysis:** Mehejabin Nurunnahar, Md Shahjahan Siraj.

**Investigation:** Mehejabin Nurunnahar, Md Shahjahan Siraj.

**Methodology:** Mehejabin Nurunnahar, Md Shahjahan Siraj.

**Supervision:** Mehejabin Nurunnahar.

**Visualization:** Mehejabin Nurunnahar, Md Shahjahan Siraj.

**Writing – original draft:** Mehejabin Nurunnahar, Tahmidul Haque, S. M. Rokonuzzaman, Susmita Dey Pinky, Rumpa Kairy, Tahrima Mohsin Mohona, Md Shahjahan Siraj.

**Writing – review & editing:** Mehejabin Nurunnahar, M. Pear Hossain, Abdus Sobhan, Most. Hafeza Khatun, Abu Yousuf Md Abdullah.

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
