## [Decision Letter · Decision Letter 0]

13 Nov 2024

PONE-D-24-27119Quality of antenatal care and its potential impacts on delivery services and postnatal care compliance among reproductive women of BangladeshPLOS ONE

Dear Dr. Nurunnahar,

Thank you for submitting your manuscript to PLOS ONE. After careful consideration, we feel that it has merit but does not fully meet PLOS ONE’s publication criteria as it currently stands. Therefore, we invite you to submit a revised version of the manuscript that addresses the points raised during the review process.

Please address the comments from both reviewers carefully before you submit a revised version of your manuscript. Ensure the revised version is written clearly. Improvements in the English language is needed throughout the manuscript. You should also carefully address any comments from the editor. Please ensure that your decision is justified on PLOS ONE’s publication criteria  and not, for example, on novelty or perceived impact.

We look forward to receiving your revised manuscript.

Kind regards,

Essa Tawfiq

Academic Editor

PLOS ONE

Journal Requirements:

Please confirm at this time whether or not your submission contains all raw data required to replicate the results of your study. Authors must share the “minimal data set” for their submission. PLOS defines the minimal data set to consist of the data required to replicate all study findings reported in the article, as well as related metadata and methods (https://journals.plos.org/plosone/s/data-availability#loc-minimal-data-set-definition ).

If your submission does not contain these data, please either upload them as Supporting Information files or deposit them to a stable, public repository and provide us with the relevant URLs, DOIs, or accession numbers. For a list of recommended repositories, please see https://journals.plos.org/plosone/s/recommended-repositories .

Reviewers' comments:

Reviewer's Responses to Questions

**Comments to the Author**

1. Is the manuscript technically sound, and do the data support the conclusions?

Reviewer #1: Partly

Reviewer #2: Partly

2. Has the statistical analysis been performed appropriately and rigorously? 

Reviewer #1: Yes

Reviewer #2: I Don't Know

3. Have the authors made all data underlying the findings in their manuscript fully available?

Reviewer #1: Yes

Reviewer #2: Yes

4. Is the manuscript presented in an intelligible fashion and written in standard English?

Reviewer #1: Yes

Reviewer #2: Yes

5. Review Comments to the Author

Reviewer #1: This manuscript examines the characteristics of women of child-bearing age who had recently given birth or were pregnant during data collection and who had accessed ANC. Data was collected in 2017 as part of the Bangladesh Health Survey. Appropriate statistical tools were used to create a dichotomous variable of quality ANC which became the outcome variable for a regression analysis.

I have the following comments before the manuscript can be considered for acceptance.

Abstract:

-Results: Please add odds ratio and 95%CI for significant variables.

Introduction:

-The authors should focus on key points. At its current form it is lengthy. The authors should reduce the length of introduction.

Methods:

if the BDHS2017 has been described elsewhere, it would be advisable to reference it and describe it more concisely. They way it is described suggests that the authors collected the data.

The data analysis part would need more details and also what cutoff was used at the

univariate stage to include variables for MV analysis or were all variables included in

the MV analysis. Ideally there is a cut-off for variables to be included in the MV.

Results:

Table: 4 Please explain why the OR for some variables (Birth order) could not be calculated.

Discussion:

Discussion needs a fair bit of work. In its current state it is repetitive, and the recommendations are too often vague. What the authors recommend from each of the key observations they have made.

Limitations section is not included: Several limitations should be included.

I propose the authors use, consult, and add the following references that is a similar study in another LMICs.

Stanikzai, M.H., Tawfiq, E., Jafari, M. et al. Contents of antenatal care services in Afghanistan: findings from the national health survey 2018. BMC Public Health 23, 2469(2023). https://doi.org/10.1186/s12889-023-17411-y

Reviewer #2: Title and Abstract

- The title is clear and informative. It may benefit from explicitly mentioning that the data is derived from the 2017 Bangladesh Demographic and Health Survey (BDHS) to emphasize the scope and rigor of the dataset used.

- Some comments provided in other sections may require addressing within the study itself. If adjustments are made, corresponding revisions in the abstract may also be necessary to ensure alignment with the study’s main findings.

Introduction

The introduction may require some restructuring for clarity. Specifically, consider the following suggestions:

- In lines 72–82, global mortality data is referenced multiple times. A more cohesive flow—from global data to South Asia, and finally to Bangladesh—would strengthen clarity and relevance. Streamlining this information could avoid redundancy.

- In lines 93–98, the eight WHO standards for quality assurance are mentioned, which may imply that these standards will be utilized in the study. Since they are not, it would be helpful to clarify their purpose here to avoid potential misunderstandings.

- The section on Focused Antenatal Care (FANC) (lines 106–117) is somewhat outdated, as the 2016 WHO guidelines now recommend a minimum of eight ANC visits to improve outcomes. Including this updated recommendation would align the background with current global standards, while also providing context for the historical use of FANC.

- Lines 112–114 suggest that reduced ANC visits (aligned with FANC) lead to reduced financial expenditures, which may inadvertently endorse lower visit frequency. This seems to contradict WHO’s updated recommendation of more frequent visits. Clarifying this point could prevent potential confusion for readers.

- In lines 125–127, ANC's impact on reducing maternal and newborn deaths is quantified using a single study reference. To strengthen this point, either generalize the impact of ANC on mortality reduction or support the statistic with additional studies.

- Lines 128–129 state, “Therefore, to properly understand the extent of reduction in MMR and NMR that could be achieved over the coming years, it is highly essential to understand the quality of ANC in Bangladesh.” This sentence could be rephrased to better connect the quality of ANC directly with maternal and neonatal mortality reductions.

- The research gap could be articulated more clearly by briefly summarizing what is already known about ANC quality in Bangladesh and identifying the specific gap this study addresses. Currently, the text alternates between discussing ANC’s role in delivery and postnatal care without clearly addressing why this gap exists in the context of Bangladesh.

- Introducing the five indicators of quality ANC earlier in the introduction, with a clear explanation, would improve clarity and set up the context for the methods and findings.

Methods

- Distinguishing between the original national survey methodology and the specific approach used in this study would enhance clarity. Briefly describing the BDHS 2017 methodology and referencing its report for further details, followed by a focused explanation of this study's specific analytical approach, would clarify the distinction.

- While I am not an expert in quantitative analysis, I recommend that another reviewer with expertise in survey methodology or statistical analysis evaluate this section for rigor and accuracy.

Results

- The demographic description of the mothers/women in BDHS 2017 at the beginning of the results section could be clarified. If the study used the full BDHS 2017 sample, rephrasing slightly to indicate that this is the sample used in the present analysis would prevent any confusion.

- The five ANC services used as indicators of "quality ANC" are main components but may not fully capture the WHO's broader definition of quality ANC. Acknowledging this limitation in interpreting quality ANC would add nuance and transparency to the findings.

- Several times, the terms "SBA delivery" and "facility delivery" are used, which may confuse readers. Given that SBA deliveries may occur outside health facilities while facility deliveries are generally attended by an SBA, a brief explanation of these terms and their distinct roles in the analysis would be helpful.

- In lines 265–267, the authors note that “a considerably higher percentage of facility delivery, delivery by an SBA, and PNC services was observed amongst mothers who had either completed their secondary education, were engaged in skilled/unskilled manual labor, belonged to the richest wealth quintile, or had a birth parity of 1.” However, the role of quality ANC in these associations is unclear. It would be beneficial to clarify how these factors interact with or are influenced by quality ANC.

- In lines 279–283, the association between demographic characteristics (e.g., education level, wealth quintile) and outcome variables such as PNC and facility delivery is discussed. It would be helpful to clarify how these relationships relate to quality ANC, as it currently appears somewhat disconnected from the main study objective.

Discussion

- The discussion could benefit from additional context on the specific challenges and barriers to providing quality ANC in Bangladesh. Including this background would enhance the relevance of the findings.

- In lines 314–323, the discussion returns to the association between education/income and delivery/PNC services. However, this seems tangential to the study’s primary focus on quality ANC. Shifting the emphasis toward findings directly related to ANC quality and maternal outcomes would make this section more cohesive and aligned with the study's objectives.

Limitations

- Adding a section on the limitations of the study regarding the measurement of quality ANC would be beneficial. Currently, only five core components of ANC are used to assess quality, which may not fully capture the comprehensive content and scope of quality ANC as defined by WHO standards. This narrower definition could limit the study’s conclusions on the true quality of ANC.

6. PLOS authors have the option to publish the peer review history of their article (what does this mean? ). If published, this will include your full peer review and any attached files.

**Do you want your identity to be public for this peer review?** For information about this choice, including consent withdrawal, please see our Privacy Policy .

Reviewer #1: **Yes: ** Muhammad Haroon Stanikzai

Reviewer #2: **Yes: ** Massoma Jafari

---

## [Author Response · Author response to Decision Letter 0]

31 Dec 2024

Response to the reviewer’s comments

Dear Reviewers,

Thank you for your constructive feedback on our manuscript. We appreciate your careful review and the valuable insights provided. In response, we have carefully revised the manuscript to address your comments and suggestions. All revisions are highlighted in the track-changes version and incorporated into the clean version of the manuscript. Authors will remain sincerely thankful if the revised manuscript can satisfy you to consider it for publication. The responses to your comments and suggestions are summarized as follows.

Title

Comment from reviewer 2: The title is clear and informative. It may benefit from explicitly mentioning that the data is derived from the 2017 Bangladesh Demographic and Health Survey (BDHS) to emphasize the scope and rigor of the dataset used.

Response: Acknowledged and addressed

Abstract

Comment from reviewer 1: Results: Please add odds ratio and 95%CI for significant variables.

Response: Risk ratio and 95%CI have been added for significant variables

Comment from reviewer 2: Some comments provided in other sections may require addressing within the study itself. If adjustments are made, corresponding revisions in the abstract may also be necessary to ensure alignment with the study’s main findings.

Response: Revisions have been made in the abstract in accordance with corresponding changes in other sections of manuscript, to keep alignment with study’s key findings.

Introduction

Comment from reviewer 1: The authors should focus on key points. At its current form it is lengthy. The authors should reduce the length of introduction.

Response: Acknowledged. Necessary revisions have been made to re-organize the flow of the introduction and also to more reflect the key points. The length of introduction also has been downsized.

Comment from reviewer 2: In lines 72–82, global mortality data is referenced multiple times. A more cohesive flow—from global data to South Asia, and finally to Bangladesh—would strengthen clarity and relevance. Streamlining this information could avoid redundancy.

Response: Acknowledged. A more cohesive flow has been tried to be established in the revised version. Also, the reference related to global mortality data represents a concrete set of evidences from 2000-2017, hence being one of the strongest found piece of literates to establish our narrative, and being the reason for multiple time referral to this document. Lines 74-80.

Comment from reviewer 2: In lines 93–98, the eight WHO standards for quality assurance are mentioned, which may imply that these standards will be utilized in the study. Since they are not, it would be helpful to clarify their purpose here to avoid potential misunderstandings.

Response: After further thorough discussion with the author group, the list of WHO standards has been dropped off in the revision, so that no misunderstanding arises.

Comment from reviewer 2: The section on Focused Antenatal Care (FANC) (lines 106–117) is somewhat outdated, as the 2016 WHO guidelines now recommend a minimum of eight ANC visits to improve outcomes. Including this updated recommendation would align the background with current global standards, while also providing context for the historical use of FANC.

Response: The discussion around Focused ANC and also recent WHO revised guidelines in 2016 has been dropped in revision for downsizing the length of introduction and also as the bringing of this topic is not a must for emphasizing upon importance of quality ANC.

Comment from reviewer 2: Lines 112–114 suggest that reduced ANC visits (aligned with FANC) lead to reduced financial expenditures, which may inadvertently endorse lower visit frequency. This seems to contradict WHO’s updated recommendation of more frequent visits. Clarifying this point could prevent potential confusion for readers.

Response: Acknowledged. The mentioned line has been dropped in revision, as the authors have not intention to endorse lower frequency of ANC visits.

Comment from reviewer 2: In lines 125–127, ANC's impact on reducing maternal and newborn deaths is quantified using a single study reference. To strengthen this point, either generalize the impact of ANC on mortality reduction or support the statistic with additional studies.

Response: Acknowledged. The mentioned line has been dropped, as there is already talk about ANC’s potential impact on PNC visit compliances, as well as maternal and newborn health outcomes, supported by multiple references in last paragraph of the introduction.

Comment from reviewer 2: Lines 128–129 state, “Therefore, to properly understand the extent of reduction in MMR and NMR that could be achieved over the coming years, it is highly essential to understand the quality of ANC in Bangladesh.” This sentence could be rephrased to better connect the quality of ANC directly with maternal and neonatal mortality reductions.

Response: Acknowledged. The sentence has been rephrased to better connect the quality of ANC with maternal and neonatal mortality reductions, and also to further clarify the motive of the study. Line 104-108.

Comment from reviewer 2: The research gap could be articulated more clearly by briefly summarizing what is already known about ANC quality in Bangladesh and identifying the specific gap this study addresses. Currently, the text alternates between discussing ANC’s role in delivery and postnatal care without clearly addressing why this gap exists in the context of Bangladesh.

Response: Acknowledged. The research gap statement has been rephrased to mention the prevalent knowledge and also the specific gap the study is going to address.

Comment from reviewer 2: Introducing the five indicators of quality ANC earlier in the introduction, with a clear explanation, would improve clarity and set up the context for the methods and findings.

Response: Acknowledged. The five indicators of quality ANC have been introduced earlier in the introduction in the revision, for further clarity of the concept.

Method

Comment from reviewer 1: If the BDHS2017 has been described elsewhere, it would be advisable to reference it and describe it more concisely. The way it is described suggests that the authors collected the data.

The data analysis part would need more details and also what cutoff was used at the

univariate stage to include variables for MV analysis or were all variables included in

the MV analysis. Ideally there is a cut-off for variables to be included in the MV.

Response: Acknowledged. The method section has been revised to focus more on process of data analysis conducted by the authors, and less on the original data collection process by BDHS. Particular cutoff for including variables for MV also has been thoroughly discussed in the revision for further convenience.

Comment from reviewer 2: Distinguishing between the original national survey methodology and the specific approach used in this study would enhance clarity. Briefly describing the BDHS 2017 methodology and referencing its report for further details, followed by a focused explanation of this study's specific analytical approach, would clarify the distinction.

Response: Acknowledged. The method section has been revised to put less focus on BDHS 2017 data collection process, and more on particular analytical approach of this study.

Results

Comment from reviewer 1: Table: 4 Please explain why the OR for some variables (Birth order) could not be calculated.

Response: The ‘Birth order’ variable was not found to fit in the adjusted model as per pre-set cutoff, hence has been dropped off from table 4 in the revision. It was mistakenly added in the first draft in table 4.

Comment from reviewer 2: The demographic description of the mothers/women in BDHS 2017 at the beginning of the results section could be clarified. If the study used the full BDHS 2017 sample, rephrasing slightly to indicate that this is the sample used in the present analysis would prevent any confusion.

Response: Acknowledged. Rephrasing has been done to clarify the sample used in the present analysis.

Comment from reviewer 2: The five ANC services used as indicators of "quality ANC" are main components but may not fully capture the WHO's broader definition of quality ANC. Acknowledging this limitation in interpreting quality ANC would add nuance and transparency to the findings.

Response: This particular limitation has been acknowledged in the ‘Strengths and limitations’ section in the revised version, to add further transparency to the findings. Lines 329-333.

Comment from reviewer 2: Several times, the terms "SBA delivery" and "facility delivery" are used, which may confuse readers. Given that SBA deliveries may occur outside health facilities while facility deliveries are generally attended by an SBA, a brief explanation of these terms and their distinct roles in the analysis would be helpful.

Response: These mentioned terms have been more elaboratively discussed in the dependent variable section of the method in revision. Lines 152-155.

Comment from reviewer 2: In lines 265–267, the authors note that “a considerably higher percentage of facility delivery, delivery by an SBA, and PNC services was observed amongst mothers who had either completed their secondary education, were engaged in skilled/unskilled manual labor, belonged to the richest wealth quintile, or had a birth parity of 1.” However, the role of quality ANC in these associations is unclear. It would be beneficial to clarify how these factors interact with or are influenced by quality ANC.

Response: This particular finding was represented in the ‘Distribution of delivery and PNC services compliances’ section, to reflect on the capacity of the mentioned population groups to have additional access to service utilization and health awareness. The idea was not to directly bind the finding with quality ANC, rather to demonstrate the particular characteristics of the population who avail facility delivery services and PNC. As we have two layered outcomes in this study, hence extrapolating the characteristics of delivery service and PNC utilizers is equally important. However, the line has been rephrased to make it more understandable.

Comment from reviewer 2: In lines 279–283, the association between demographic characteristics (e.g., education level, wealth quintile) and outcome variables such as PNC and facility delivery is discussed. It would be helpful to clarify how these relationships relate to quality ANC, as it currently appears somewhat disconnected from the main study objective.

Response: Acknowledged. Explanation has been added in the revision to clarify how these relationships are also related to quality ANC.

Discussions

Comment from reviewer 1: Discussion needs a fair bit of work. In its current state it is repetitive, and the recommendations are too often vague. What the authors recommend from each of the key observations they have made.

Response: Acknowledged. The discussion section has been revised to avoid repetition, and to further consolidate recommendations.

Comment from reviewer 1: Limitations section is not included: Several limitations should be included.

Response: A ‘Strengths and limitations’ section has been added at the end of discussion.

Comment from reviewer 1: I propose the authors use, consult, and add the following references that is a similar study in another LMICs.

Stanikzai, M.H., Tawfiq, E., Jafari, M. et al. Contents of antenatal care services in Afghanistan: findings from the national health survey 2018. BMC Public Health 23, 2469(2023). https://doi.org/10.1186/s12889-023-17411-y

Response: The particular reference suggested by the reviewer has been added for comparing with findings from a similar study. Lines 299-302.

Comment from reviewer 2: The discussion could benefit from additional context on the specific challenges and barriers to providing quality ANC in Bangladesh. Including this background would enhance the relevance of the findings.

Response: Acknowledged. Additional contexts have been added on specific challenges and barriers that hinder providing quality ANC in Bangladesh. Lines 286-290.

Comment from reviewer 2: In lines 314–323, the discussion returns to the association between education/income and delivery/PNC services. However, this seems tangential to the study’s primary focus on quality ANC. Shifting the emphasis toward findings directly related to ANC quality and maternal outcomes would make this section more cohesive and aligned with the study's objectives.

Response: Maternal health outcomes are substantially influenced via facility delivery and PNC services. Hence these confounders have been discussed in the mentioned lines. However, additional explanations have also been made in the revision to keep alignment of these findings with study’s objectives and also to mark how these findings are relatable to ANC quality. Lines 312-316.

Comment from reviewer 2: Adding a section on the limitations of the study regarding the measurement of quality ANC would be beneficial. Currently, only five core components of ANC are used to assess quality, which may not fully capture the comprehensive content and scope of quality ANC as defined by WHO standards. This narrower definition could limit the study’s conclusions on the true quality of ANC.

Response: A ‘strengths and limitations’ section has been added at the end of discussion. Also, the particular limitation about BDHS definition of quality ANC around fiver core components not fully capturing the comprehensiveness of quality ANC as per WHO standards has been acknowledged. Lines 318-333.

Response to the Editorial Comments:

Journal Requirements:

Response: We have carefully reviewed the PLOS ONE style guidelines and ensured that our revised manuscript adheres to the journal's formatting and submission requirements.

Response: We have ensured that all raw data required to replicate the findings of our study is provided in accordance with PLOS ONE's data sharing policies. The data S2 file and supporting S1 Table have been uploaded as Supporting Information, and referenced in the manuscript text accordingly.

3. If there are ethical or legal restrictions on sharing a de-identified data set, please explain them in detail (e.g., data contain potentially sensitive information, data are owned by a third-party organization, etc.) and who has imposed them (e.g., an ethics committee). Please also provide contact information for a data access committee, ethics committee, or other institutional bo

---

## [Decision Letter · Decision Letter 1]

14 Mar 2025

Quality of antenatal care and its potential impacts on delivery services and postnatal care compliance among reproductive women in Bangladesh: a situation analysis from the Bangladesh Demographic and Health Survey 2017

PONE-D-24-27119R1

Dear Dr.Nurunnahar,

We’re pleased to inform you that your manuscript has been judged scientifically suitable for publication and will be formally accepted for publication once it meets all outstanding technical requirements.

Kind regards,

Essa Tawfiq

Academic Editor

PLOS ONE

---

## [Editor Report · Acceptance letter]

PONE-D-24-27119R1

PLOS ONE

Dear Dr. Nurunnahar,

I'm pleased to inform you that your manuscript has been deemed suitable for publication in PLOS ONE. Congratulations! Your manuscript is now being handed over to our production team.

Kind regards,

on behalf of

Dr. Essa Tawfiq

Academic Editor

PLOS ONE